# Arteriovenous Hemodialysis Access Stenosis Diagnosed by Duplex Doppler Ultrasonography: A Review

**DOI:** 10.3390/diagnostics12081979

**Published:** 2022-08-16

**Authors:** Jan Malik, Cora de Bont, Anna Valerianova, Zdislava Krupickova, Ludmila Novakova

**Affiliations:** 13rd Department of Internal Medicine, General University Hospital, U Nemocnice 2, 12808 Prague, Czech Republic; 2Center for Vascular Access, First Faculty of Medicine of General University Hospital, Charles University, 12108 Prague, Czech Republic; 3Vascular Laboratory, Bravis Hospital, 4624 VT Bergen op Zoom, The Netherlands; 4Faculty of Mechanical Engineering, Jan Evangelista Purkyne University, 40096 Usti nad Labem, Czech Republic

**Keywords:** arteriovenous fistula, arteriovenous graft, hemodialysis, vascular ultrasonography

## Abstract

Arteriovenous fistula (AVF) is currently the hemodialysis access with the longest life expectations for the patients. However, even the AVF is at risk for many complications, especially the development of stenosis. The latter can not only lead to inadequate hemodialysis but also lead to AVF thrombosis. Duplex Doppler ultrasonography is a very precise method, in the hands of experienced professionals, for the diagnosis of AVF complications. In this review, we summarize the ultrasound diagnostic criteria of significant stenoses and their indication for procedural therapy.

## 1. Introduction

Hemodialysis, a renal function replacement method, needs a regular entry into the bloodstream with sufficient blood flow (at least 500 mL/min). The insertion of a special central vein catheter, introduced into the superior or inferior vena cava, is an option. However, the long-term use of hemodialysis catheters is associated with a significantly increased risk of infection and mortality [1]. The preferred vascular access is an arteriovenous fistula (AVF), which is the direct connection of a patient’s artery and vein, or an arteriovenous graft (AVG), which is when an artificial (polytetrafluoroethylene) vascular graft is used for punctures in cases of abandoned superficial veins [2].

Both AVF and AVG bypass the resistant arterioles. Therefore, the flow volume through the extremity with the access is >10 times after AVF/AVG creation. Although AVFs are considered the best currently available hemodialysis access (over AVGs and catheters), their lifespan is limited by several complications, of which stenoses are by far the most frequent [3]. They lead to decreased flow through the AVF/AVG. Thrombus formation is accelerated, and access lifespan is in danger. Duplex Doppler ultrasonography (DUS) is a very sensitive and precise method for assessing AVF/AVG stenoses [3]. However, published articles differed significantly in stenosis definition criteria, as we show below.

The power of DUS to detect significant access stenosis, which could, in turn, lead to access thrombosis, was confirmed already in the early nineties [4,5]: >50% of stenosis is associated with a >50% risk of access thrombosis within 6 months. Currently, there is no doubt that DUS can diagnose access stenosis, and the measurement of residual diameter is very precise in comparison with angiography [6]. Moreover, the advantage over angiography is that it is more precise in the explanation of stenosis etiology (e.g., intimal hyperplasia, post-puncture scarring, and pressure of surrounding structures), the monitoring of access maturation, and the calculation of access flow volume. 

In this review, we describe ultrasound diagnostics of AVF/AVG stenoses.

## 2. General Definition of Stenoses

Vascular narrowing of any etiology results in local hemodynamic and structural disturbances. Hemodynamic changes include flow acceleration, changes in wall shear stress, pressure drop, and decrease in flow volume. Flow velocity increase assessment in the stenotic segment or proximal to it is a part of stenosis quantification by DUS. Hemodynamic changes due to a stenosis are depicted in Figure 1.

AVF flow volume (Qa = amount of blood flowing over time, expressed as mL/min) depends on the perfusion pressure (the mean arterial pressure minus the central venous pressure) and on access resistance. The latter is determined mainly by the size of the arteriovenous anastomosis, but in stenotic AVF/AVGs, the stenosis increases access resistance (and thus decreases flow volume) more significantly [7]. It is particularly true for juxta-anastomotic or arterial stenoses.

## 3. Etiology and Types of Stenoses

Several etiologies play a role in the development of access stenoses.

Atherosclerosis and/or medial calcinosis are the most frequent causes of arterial narrowing. The former is represented by focal stenosis due to an atherosclerotic plaque; the latter affects diffusely, especially in forearm arteries. 

Surgical manipulation and trauma contribute to the development of juxta-anastomotic stenoses. 

Intimal or intimomedial hyperplasia is the most common etiology of AVF/AVG stenoses (Figure 2). In AVFs, it typically develops in the outflow vein within a few centimetres from the anastomosis and, sometimes, the cephalic arch is affected. Many mechanisms are suspected to be the cause, including trauma and wall shear stress alterations. In AVGs, intimal hyperplasia develops most frequently in the venous anastomosis, adjacent outflow vein, or more proximally in the basilic vein. The inability of the vein to dilate is another etiology that contributes to stenosis formation in the juxta-anastomotic region and cephalic arch. Other mechanisms include healed thrombophlebitis, thickened venous valves, and outer compression by a hematoma, seroma, cervical ribs, dilated aortic arch, etc., and by a pseudoaneurysm (Figure 3). The long-term placement of central venous catheters is a known risk factor for central vein stenosis.

Non-occluding thrombus could also cause a stenosis (Figure 4).

## 4. Predilectional Sites of Stenoses

The typical location of arteriovenous access stenosis is different for AVFs and AVGs (Figure 5). It also depends on the location of the access.

In native radiocephalic fistulas, stenoses typically occur in the juxta-anastomotic segment of the outflow vein (Figure 5) [8]. Suggested risk factors for stenosis formation in this site include loss of vasa vasorum during the surgical procedure, changes in shear stress, or increased blood flow turbulence [9,10]. The typical site for brachiocephalic fistula is represented by cephalic arch stenosis [8] that could be caused by a compression of the vein by the clavipectoral fascia, the sharp angle of the cephalic vein, or the higher frequency of valves causing turbulent flow [11]. Proximal swing stenosis is specific for brachiobasilic fistula, caused by kinking or compression in the artificial tunnel, constriction by fascia, or by turbulent flow caused by angulation of the vein [8,12]. According to the retrospective study by Badero et al. [9], proximal swing segment stenoses and juxta-anastomotic stenoses are dominant in native AVFs, followed by puncture site stenoses.

The most typical location of AVG stenosis is the venous anastomosis [13] and the adjacent segment of the outflow vein, probably due to altered haemodynamic conditions [14]. Other sites—in-graft stenoses, outflow vein stenoses, or stenoses of arterial anastomosis—are less frequent. Central vein stenoses are usually linked to the previous use of hemodialysis catheters [15]. Stenoses of the feeding artery are present only in a minority of AV accesses [16]. Their etiology includes atherosclerosis or medial calcinosis.

The feeding artery (including the subclavian artery) could be affected by atherosclerosis and/or medial calcinosis. Atherosclerosis is relatively rare in the upper extremity arteries but more frequent in the subclavian or innominate artery. The latter must be suspected when low Qa has no clear explanation (Figure 6). Severe subclavian artery stenosis or occlusion could also be visualized indirectly by the flow reversal in the ipsilateral vertebral artery. While atherosclerosis affects the intimal layer of the arterial wall, medial calcinosis affects the medial (muscle) layer and is typical, particularly for patients with advanced type 2 diabetes mellitus. Medial calcinosis can be specifically observed in the forearm (and hand) arteries and is characterized by a diffuse arterial narrowing and hyperechoic arterial wall. It could be responsible for slow AVF maturation, low Qa, and/or hand ischemia [17].

## 5. Significant Stenosis Definition by Ultrasonography

DUS is a method that is comparable to and, in some respects, better than angiography. The advantages of DUS over angiography not only include the noninvasive visualization of stenosis etiology and of surrounding tissues but also the calculation of access flow volume (Qa). Angiography determines the significance of a stenosis, usually by a luminal diameter reduction of ≥50% (associated with clinical and/or physiological abnormalities). 

Traditionally, DUS determined AVF/AVG stenosis by the parameters that are used in other vascular beds. However, vascular access differs considerably from natural arteries and veins. Finding the thresholds of these traditional criteria to define which stenosis is significant in vascular access seems to be challenging. Studies that made attempts had their flaws, e.g., retrospective, selection bias, small sample size, DUS and angiography not performed at the same time, and the lack of including a control group [18,19,20,21]. Although these studies did agree on certain points, they were not uniform and had contradictive findings. Currently, stenosis criteria used by different authors include diameter/cross-sectional area reduction, the quantification of flow acceleration by the stenosis, and the assessment of Qa and residual diameter. Strict diagnostic criteria for a significant stenosis are of particular importance especially because the outflow vein diameter is often irregular and the unnecessary treatment of a non-significant stenosis brings more pain to the patient, higher costs and the quicker development of restenosis than the progression of an untreated and non-significant stenosis [22]. 

Stenosis leads to flow acceleration to maintain the transportation of the mass of fluid. DUS determined the increase in the peak systolic velocity (PSV) inside the stenotic segment and/or its ratio to the prestenotic segment. These parameters were introduced at the beginning of DUS, in which the low resolution made it impossible to grade stenosis and occlusion solely upon gray-scale images. The PSV ratio of 2.0 to 2.9 presents stenosis as 50% to 74%, and a PSV ratio of more than 3.0 presents stenosis as more than 75% [21]; therefore, the used cut-off for a significant stenosis is a PSV ratio of >2–3. In fact, flow acceleration inside a stenosis represents a drop in blood pressure (according to the Bernoulli equation). The disadvantages of using flow acceleration include natural accelerations in curved/kinked parts of the vascular tree.

Percent stenosis determination depends on what is being defined as the normal reference part of the vascular access, which is a weakness that DUS shares with angiography. After the access creation, the vein undergoes a nonuniform dilatation. Various levels of dilatation could be found at several points in the access, e.g., vessel curves, venous valves, cannulation sides, bifurcations, and the juxta-anastomotic side [23,24]. Hence, defining the normal reference value is challenging, and percent stenosis determination is fraught with uncertainty [6,23,24]. Despite these limitations, the concept of “stenosis > 50%” is still recommended in scientific articles [25]. 

The measurement of the residual or minimal lumen diameter (Figure 7), meaning the absolute minimal lumen diameter of the narrowest point within the stenosis (lesion), was introduced as a parameter to discriminate between significant and non-significant stenoses [6,26]. Studies have shown that the residual diameter is an accurate parameter in distinguishing functional from dysfunctional fistulas and has a superior correlation with the Qa when compared with a simple diameter reduction of >50% [27,28]. However, these studies differed in the used markers by which the dysfunctionality was determined. For example, one study found a cut-off value for RD of ≤2.7 mm to be representative of a dysfunctional radiocephalic fistula, but after including the value of the Qa, this cut-off changed towards a value of ≥3.2 mm for a functional fistula [29]. Residual diameter cut-off values for a significant stenosis differed between 1.9 and 4.5 mm in various studies [6,26,28]. We believe that higher values of residual diameter are not ideal because we frequently find well-functioning AVFs with 3.0 mm wide radial arteries or arteriovenous anastomoses. There is an ongoing debate about this topic.

The flow volume through the vascular access (Qa) is considered to be a functional marker. It can be measured during hemodialysis by dilution techniques and at any time by DUS. The principle of the Qa calculation by ultrasonography is shown in Figure 8. In native AVFs, Qa is usually measured in the brachial artery. This is because the outflow vein frequently has an irregular diameter, side branches, and is easily compressible by the ultrasound probe, which changes its circular cross-section to a parabolic one. When the cross-section changes to parabolic, the equation mentioned below is not valid anymore. In patients with arteriovenous grafts, Qa is measured directly in the graft and in a straight segment close to the venous anastomosis. However, some centers measure Qa directly in the outflow vein [30]. Measuring Qa at least three times and using the average value are advised. The sample volume should encompass the entire vessel’s diameter to include the low-velocity layers present along the vessel wall. Measurement in a straight vascular segment is mandatory to assure the quantification of a laminar flow. Usual Qa values reach 500–1500 mL/min in AVFs and 600–1500 mL/min in grafts. Low values bring the risk of inadequate hemodialysis and access thrombosis. Higher values could be associated with heart failure or hand ischemia [31]. Importantly, Qa physiologically fluctuates with blood pressure and hydration levels—as does the cardiac output [32].

Complex criteria: The Prague group has defined complex DUS criteria, which include two main parameters (percentage of diameter reduction and the value of flow acceleration) and additional ones [22]. They are described in Table 1. In these criteria, significance is based upon anatomical, functional, and hemodynamical parameters, which are regarded as complementary. These criteria also enable one to define a stenosis that has the potential to progress and needs closer and more frequent observation, so-called a borderline stenosis [33]. Although the aforementioned criteria [22] were originally designed for AVGs, research has shown that they can be used to define the severity of the stenosis in arteriovenous fistulas [34].

## 6. Arteriovenous Access Surveillance by Ultrasonography

Surveillance is the term used for the regular assessment of AVF/AVG by DUS. Given the advances of DUS, which include precise non-invasive visualization, including the quantification of the flow volume, the idea of performing DUS surveillance seems to be very attractive. Its philosophy is based on the regular ultrasound AVF/AVG assessment with preemptive intervention (usually by percutaneous balloon angioplasty) when a significant stenosis is present. Considering surveillance techniques, the higher is the risk of a stenosis in the access, the higher the benefit of surveillance. AVFs have the longest long-term patency, but up to 50% of AVFs never mature [35]. AVF maturation could be facilitated by PTA [36], and DUS can be effectively used for depicting the outflow vein and its stenosis (Figure 9). AVGs have shorter patency than AVFs, especially due to significantly more frequent stenosis development by intimomedial hyperplasia, especially in the venous anastomosis or adjacent outflow vein [37]. Shortly, surveillance could be more beneficial in AVGs, and indeed, most of the surveillance trials studied AVGs. Various randomized controlled trials analyzed the benefit of AVG surveillance [38,39,40,41,42]. Their findings were conflicting. However, the trials differed in the number of patients (the largest study included 192 subjects) by including incident or prevalent patients in the treatment methods (PTA vs. surgery), but particularly in the criteria of stenosis significance. While some authors used only the combination of lumen narrowing in B-mode + flow acceleration, others used more detailed complex criteria shown in Table 1 but with different cut-off values of the residual diameter. Unfortunately, two meta-analyses [43,44] put together all of these conflicting and heterogenous data and concluded that ultrasound surveillance does not bring benefit to the patients. As a result, instead of performing a larger multicentric trial and testing its benefits, currently, surveillance is not generally supported for AVGs and is considered doubtful for AVFs [45,46].

## 7. Symptoms of AVF/AVG Stenosis and When to Intervene a Stenosis

A regular physical examination of AVF/AVGs is recommended and supported by the KDOQI guidelines [46]. Clinical indicators (signs and symptoms) suggesting clinically significant stenoses are summarized in Table 2. The guidelines recommend the treatment of only symptomatic stenoses. 

The stenosis of the feeding artery, anastomosis, and juxta-anastomotic part of the outflow vein usually lead to Qa reduction and higher recirculation of the dialyzed blood. This could be manifested by an easier collapsibility of the outflow vein during extremity elevation. Low Qa could lead to inadequate dialysis dose and excessive collapse of the outflow veins during arm elevation. Inadequate dialysis dose (Kt/V) could be a result of low Qa. 

Cannulation problems are not rare but should be new in the patient to suspect a stenosis. Stenosis in the puncture area or low Qa could be responsible. Other explanations include situations when the outflow vein is diffusely narrow, deeply located (>6 mm from the skin surface), branched, etc.

Venous hypertension involves, i.e., the failure of outflow vein collapse during arm elevation or increased dynamic venous pressure, or prolonged bleeding from the puncture site that develops in proximal outflow vein stenosis, usually in the arm and also in the central veins. On the contrary, central vein stenosis seldom leads to decreased AVF flow volume and should not be treated if asymptomatic [46,47]. However, they are frequently missed by the ultrasound because the central veins are hidden behind bone structures.

Analysis of the symptoms relies on the experience of the hemodialysis staff and the available methods, such as Qa measurement by dilution methods. The indication for a preemptive balloon angioplasty has to be considered with caution, ideally with the cooperation between the dialysis staff, interventionalist, and ultrasonographer. The latter could contribute by using the precise criteria of a significant stenosis and the complex criteria mentioned above as an example. Importantly, the Doppler ultrasound is generally recommended in cases of any problem or uncertainty related to vascular access.

## 8. Conclusions

DUS is a non-invasive, cheap, and in educated and experienced hands, also a very precise method for diagnosing AVF/AVG stenoses. Its goal is to diagnose stenoses that cause clinical problems, choose an appropriate puncture site for percutaneous therapy, and identify stenoses that possess a high risk of access thrombosis without unnecessary interventions. The latter goal is, however, the most conflicting, and this is probably due to the nonuniform criteria of significant stenoses used in published articles. Therefore, the current general recommendation by the guidelines supports the treatment of only symptomatic stenoses [45,46]. Unfortunately, literary data on the role of physical examination in diagnosing significant AVF/AVG stenoses are conflicting [48,49,50,51,52,53]. We believe that this fact, together with the growing trust of the younger generation in instrumental methods, would further widen ultrasound diagnostics of AVF/AVG complications. However, this cannot be achieved without quality education and appropriate training in vascular access ultrasonography. 

## Figures and Tables

**Figure 1 diagnostics-12-01979-f001:**
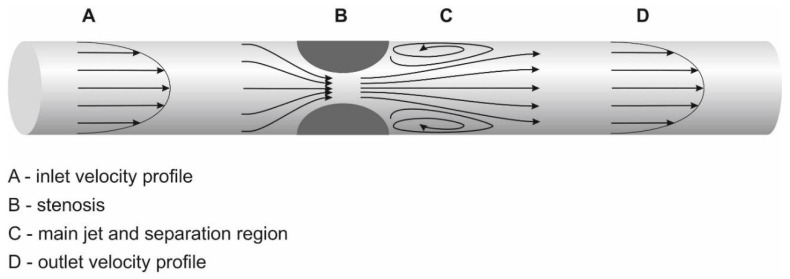
**Arteriovenous fistula with a stenosis: hemodynamic implications.**Legend: vascular segments are marked A–D. A = laminar flow; B = stenosis with flow acceleration; C = region of separation with reversed flow; D = area of flow reattachment. A significant stenosis causes blood pressure to drop, i.e., blood pressure in segment “A” is higher than in “B”. Flow acceleration depends on stenotic severity and on the geometry (asymmetric stenosis causes a higher pressure drop than symmetric stenosis). Flow separation that occurs just distal to the stenosis is responsible for the pressure drop but also narrows the flow jet even more significantly than the stenosis itself. Laminar flow is associated with physiological values of wall shear stress (WSS). WSS, however, increases inside the stenosis, where they can denudate endothelium and activate the von Willebrand factor. WSS has a changing vector in the segment “C”, which contributes to intimal hyperplasia development.

**Figure 2 diagnostics-12-01979-f002:**
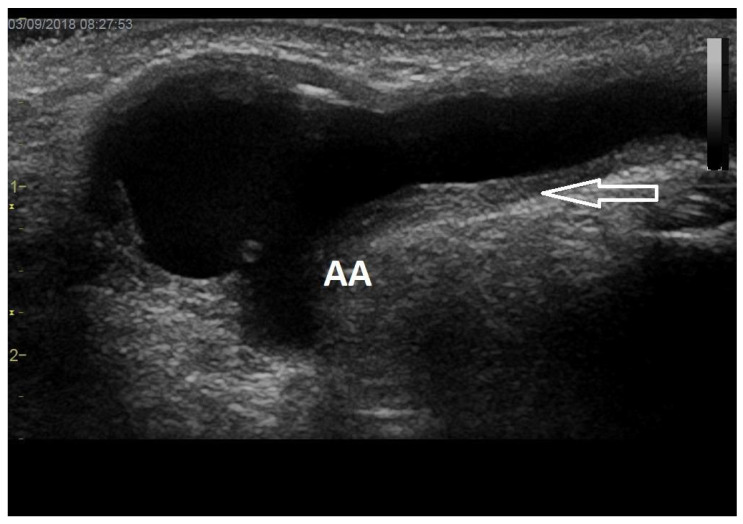
**Juxta-anastomotic outflow vein stenosis.**Legend: Juxta-anastomotic intimal hyperplasia (arrow) is visible here as the thickening of the venous wall leads to lumen narrowing. This is the most frequent etiology of AVF/AVG stenosis. AA stands for arterial anastomosis.

**Figure 3 diagnostics-12-01979-f003:**
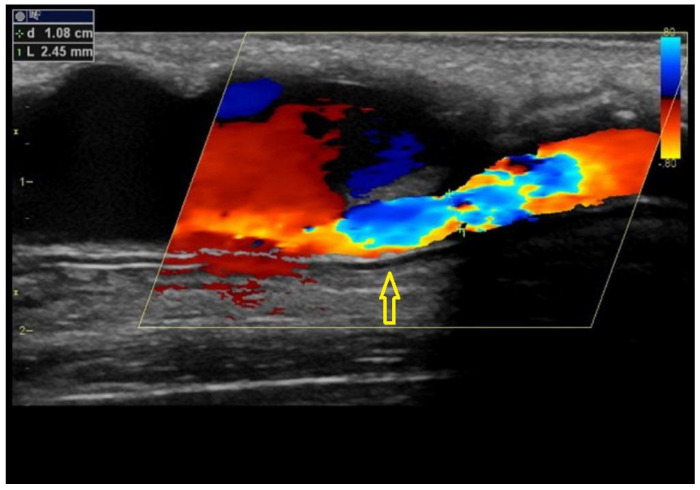
**Pseudoaneurysm of an arteriovenous graft causing stenosis.**Legend: Arteriovenous graft is affected by a large pseudoaneurysm, part of which compresses the graft itself (arrow). An unaffected part of the graft is on the right side. This is a less frequent etiology of AVG stenosis.

**Figure 4 diagnostics-12-01979-f004:**
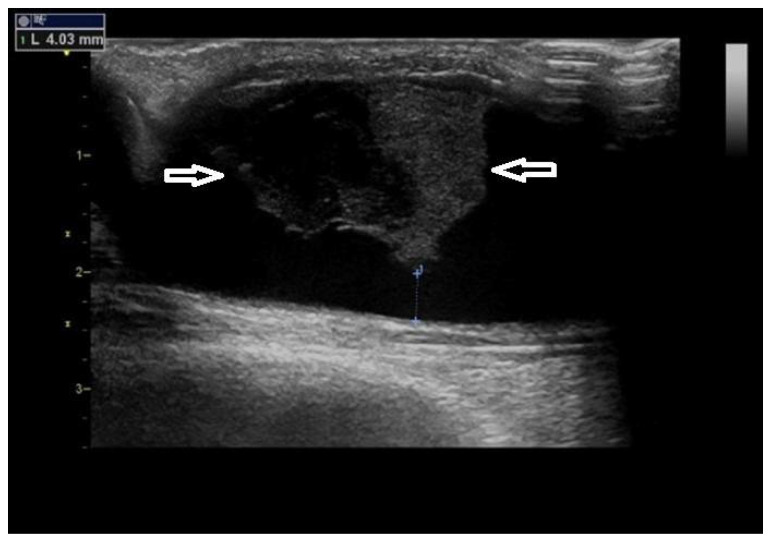
**Thrombus in the outflow vein.**Legend: A thrombus (arrows) in a dilated outflow vein. Acute thrombi are hypo-echoic (darker), as is the left part of this thrombus. The resulting stenosis is not significant (residual diameter 4 mm), albeit the thrombus could further progress. This one was dissolved by systemic anticoagulation therapy.

**Figure 5 diagnostics-12-01979-f005:**
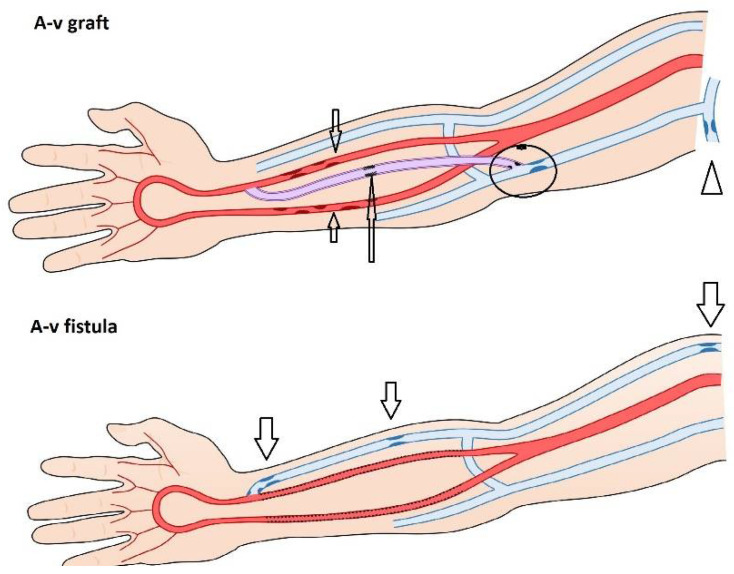
**Typical sites of arteriovenous fistula or graft stenoses (arrows).**Legend: Medial calcinosis of the forearm arteries may affect both AVF/AVGs. Native AVFs stenoses develop most frequently in the outflow vein just proximal to the anastomosis in the puncture area or more centrally—in the cephalic arch. In AVGs, stenoses may affect the venous anastomosis, the adjacent outflow vein segment, or the outflow vein more proximally. Again, repeated punctures into the same site could lead to graft stenosis induced by healing and scar formation.

**Figure 6 diagnostics-12-01979-f006:**
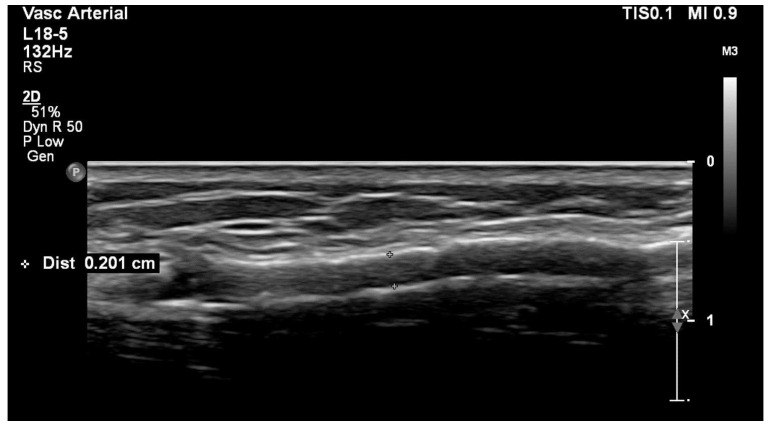
**Medial calcinosis of the radial artery feeding a radiocephalic fistula.**Legend: Longitudinal section with the use of a high-resolution probe. Hyper-echoic (white) structures are in the arterial wall and represent calcifications in the medial layer.

**Figure 7 diagnostics-12-01979-f007:**
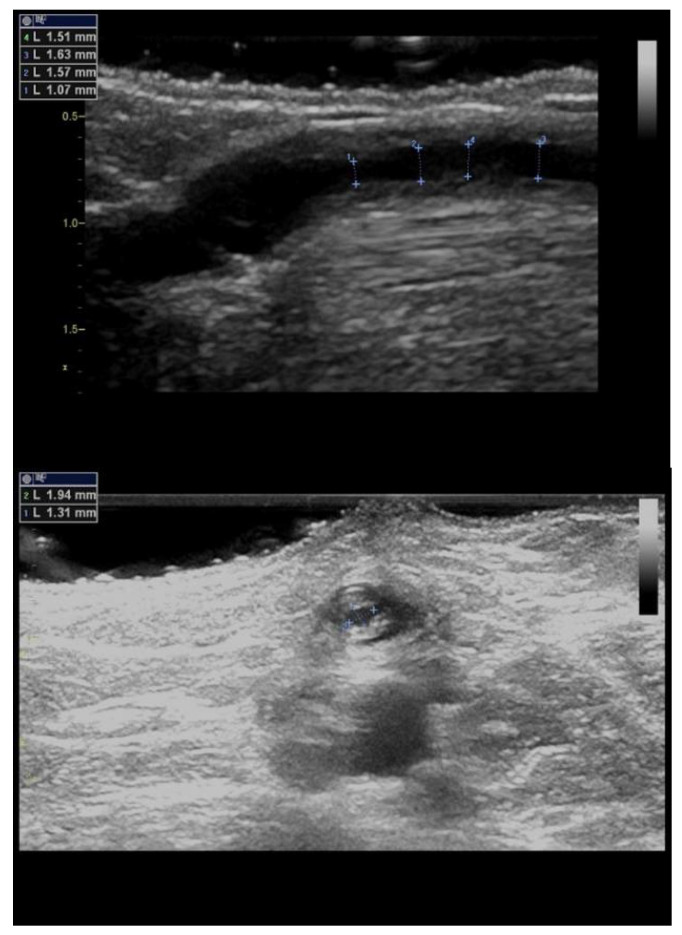
**Residual diameter measurement by ultrasound.**Legend: Longitudinal and cross-sectional views of stenoses caused by intimal hyperplasia. Small crosses represent caliper and the residual diameter value is shown on the upper left corner.

**Figure 8 diagnostics-12-01979-f008:**
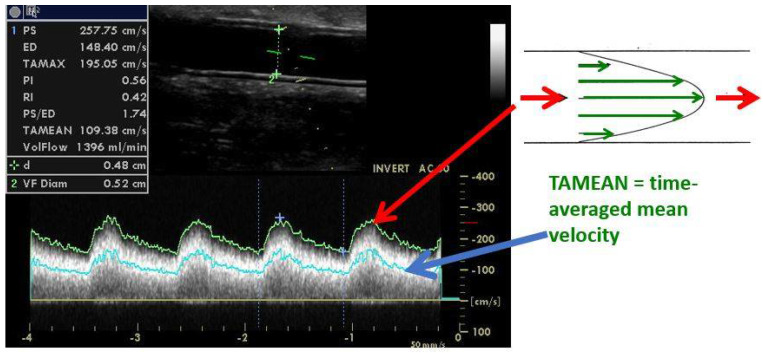
**Flow volume calculation by ultrasound.**Legend: AVF flow volume (Qa) is measured in the brachial artery, while AVG Qa is measured directly in the graft, as in this example. The calculation is based on the cross-sectional area (πr^2^, where r = radius) and time-averaged mean velocity (TAMEAN), which represents the mean velocity during a cardiac cycle. When the velocity is in cm/s, and the radius or diameter is in cm, the equation: *Qa = πr^2^ × TAMEAN × 60* would give flow volume in mL/min. The same Doppler angle should be used for comparisons (usually 60° as in this case).

**Figure 9 diagnostics-12-01979-f009:**
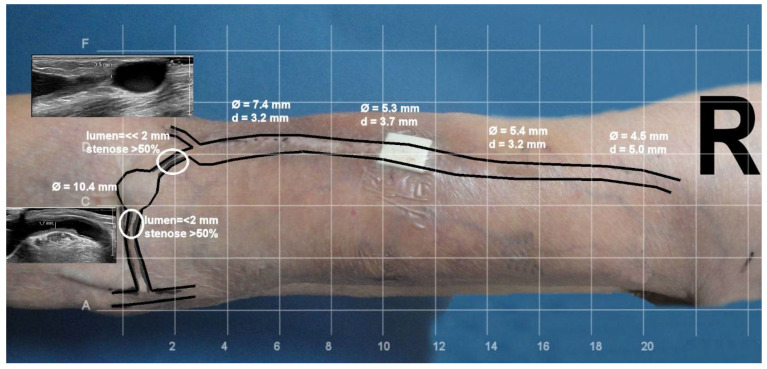
**Depiction of a brachiocephalic fistula with the use of ultrasonography.**Legend: Drawing venous segments on the patient’s skin is not only of value for hemodialysis nurses but also for interventionalists prior to a procedure. Both the depth from the skin surface (d) and venous diameter (φ) are presented.

**Table 1 diagnostics-12-01979-t001:** Complex ultrasound criteria of a significant vs. borderline stenosis.

SIGNIFICANT	BORDERLINE
**Main criteria**	
Diameter reduction by >50%	
Peak systolic velocity increase > 2–3x	
**+Additional criteria (≥1)**	
Residual diameter < 1.9–2.0 mm	No additional criterion
Flow volume decrease by >25% *	
Flow volume < 600 mL/min for AVGs, <500 mL/min for AVFs	

Legend: Two main criteria and at least one additional criterion characterized a significant stenosis. If only 1–2 main criteria are present, the stenosis is borderline, and re-evaluation is indicated within 6–8 weeks. Significant stenoses are indicated to correction. * Flow volume decrease by >25% if the previous value was <1000 mL/min. Stenoses characterized by none or only 1 main criterion are considered non-significant.

**Table 2 diagnostics-12-01979-t002:** Clinical indicators suggesting clinically significant AVF/AVG lesion.

**Physical examination**	Ipsilateral extremity edema
	Pulse alterations (weak or resistant pulse), difficult to compress in the area of stenosis
	Abnormal thrill (weak, discontinuous) with only the systolic component in the stenotic area
	Abnormal bruit (high pitched with a systolic component in the area of stenosis)
	Failure of outflow vein collapse during arm elevation
	Lack of pulse augmentation during arm elevation
	Excessive collapse of the outflow veins during arm elevation
**During hemodialysis**	New difficulty with cannulation
	Aspiration of blood clots
	Inability to achieve the target dialysis blood flow
	Prolonged bleeding after needle withdrawal for 3 consecutive dialysis sessions
	Unexplained decrease in the target dialysis dose (Kt/V) on a constant dialysis prescription and without dialysis prolongation

## Data Availability

Not applicable.

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
