# Peer review of "Arteriovenous Hemodialysis Access Stenosis Diagnosed by Duplex Doppler Ultrasonography: A Review"

_diagnostics, 2022, doi:10.3390/diagnostics12081979_

Round 1

Reviewer 1 Report

The authors have prepared a nice overview on the use of US in the diagnostics of stenosis in the vascular access. I only have a few comments:

-the title of the manuscripts says it will focus on the diagnostic criteria for (significant) stenosis, but the manuscript also discusses other complications, therefore either the title should be changed, or better, the section on "other complications" should be removed.

- the last section on "symptoms of stenosis" is too general and discusses other issues also (indications for US exam), please focus more on the clinical signs that indicate the stenosis is significant

- in some parts, english should be improved:

* pg 4,  paragraph 4: what is "intimohyperplasia"?

* the first sentence of the conclusions

Author Response

Dear Reviewer,

Thank you for your valuable help. We have followed your suggestions and oriented only at the stenoses. The changes are highlighted by red.

Sincerely, Jan Malik

Reviewer 2 Report

In the text, the authors discuss the role and usefulness of Doppler ultrasound in diagnosis of dialysis vascular access stenosis. The issue is not new and has been widely discussed for many years. Role of ultrasound was extensively reviewed in the recent special issue of JVA. Ultrasound is routinely done by interventionalists performing vascular access procedure. It is also routinely done as a first diagnostic step in case od vascular access dysfunction. In the chapter related to intervention clinical relevance of stenosis is not mentioned. To sum up, the text does not add much new to the current state of knowledge. 

Author Response

Dear Reviewer,

Thank you for your comments. According to the other Reviewer´s suggestion, we oriented more on stenoses. We still believe that this topic is not fully understood by the ultrasonographers, mainly becaused various authors used different criteria.

Sincerely, Jan Malik

Round 2

Reviewer 2 Report

Changes done by the authors significantly improved the manuscript and pointed out necessity of treatment of clinically important stenosis. My major remark related to novelty of the issue persists but I agree that problems of vascular access are not be fully understood by all ultrasonographers, but also other specialists. Let me add some minor comments:

1. The value of physical examination relays on experience. If PE is done by radiologists (ref. 48)  its quality is probably worse compared to nephrologists. Another data support more PE (PMID: 17928468; 21454718; 18248519). 

2. I suggest to move a part of conclusions to previous paragraph. My proposal is "DUS is a non-invasive, cheap and in experienced hands also a very precise method for diagnosing AVF/AVG complications, especially stenoses.  To avoid unnecessary interventions only clinically significant stenoses should be treated."

3. The sentence related residual diameter of 3 mm needs to be rephrased. My personal experience supports the limit at about 3 mm.

4. In paragraph "Symptoms of AVF/AVG stenosis and when to intervene a stenosis" please change lesions to stenosis in the second sentence. I suggest to add sentence promoting performance of Doppler ultrasound in case of any problem related to vascular access (high level of vigilance).

5. Short paragraph about training in vascular access ultrasonography would be of great importance. 

Author Response

Dear Reviewer,

We thank you for your important remarks and for the time you spent with reading our manuscript.

We believe that using different cut-off values of stenosis residual diameter between you and us is a good example documenting that manuscripts about this topic should be further written.

Our responses follow your suggestions:

  1. The value of physical examination relays on experience. If PE is done by radiologists (ref. 48)  its quality is probably worse compared to nephrologists. Another data support more PE (PMID: 17928468; 21454718; 18248519). We have added references that support PE as you suggested and rephrased the sentence. We did not mention radiologist vs. nephrologist because in the study No. 49 PE was done by nephrologists.
  2. I suggest to move a part of conclusions to previous paragraph. My proposal is "DUS is a non-invasive, cheap and in experienced hands also a very precise method for diagnosing AVF/AVG complications, especially stenoses.  To avoid unnecessary interventions only clinically significant stenoses should be treated." We have re-written the conclusions.
  3. The sentence related residual diameter of 3 mm needs to be rephrased. My personal experience supports the limit at about 3 mm. Our cut-off 2.0 mm comes from our trials showing that stenosis with residual diameter >2.0 mm practically do not clot (for example ref. 33). However, we have “soften” our sentences according to your suggestion.
  4. In paragraph "Symptoms of AVF/AVG stenosis and when to intervene a stenosis" please change lesions to stenosis in the second sentence. I suggest to add sentence promoting performance of Doppler ultrasound in case of any problem related to vascular access (high level of vigilance). Changed according to your suggestion.
  5. Short paragraph about training in vascular access ultrasonography would be of great importance. We agree with you and have added a sentence to the Conclusions.

Sincerely,

Jan Malik on behalf of all coauthors